# The Role of the Master Plan in City Development, Latakia Master Plan in an International Context

Nebras Khadour, Albert Fekete * and Máté Sárospataki *

Institute of Landscape Architecture, Urban Planning and Garden Art, Hungarian University of Agriculture and Life Sciences—MATE, 1118 Budapest, Hungary; nebras.khadour@phd.uni-mate.hu
* Correspondence: fekete.albert@uni-mate.hu (A.F.); sarospataki.mate@uni-mate.hu (M.S.)

**Abstract:** The master plan has been a critical instrument for shaping the development of cities worldwide. This article delves into the impact of a well-designed master plan on shaping and transforming the structure of a city, while also exploring the various aspects that can be adapted in different contexts and conditions. The article aims to highlight how an effective master plan can drive development, guide urban growth, and offer a comprehensive framework for decision-making. Specifically, this study analyses the Latakia (SY) master plan, which was proposed in 2008, and compares it with the master plans of Barcelona (ES) and Montpellier (FR), two cities with significant experience in master planning. The analysis was conducted using several key criteria, such as general vision, housing policies, urban mobility, and green network. The results showed that urban development strategies in the Latakia master plan were of limited efficiency range compared to the other case studies, as it focused on tourism and economic development rather than providing an approach for sustainable city development. Therefore, this study recommends revising the development strategies of the Latakia master plan and addressing its limitations to improve the city's structure, increase its sustainability, and quality of life. This article contributes to the existing body of knowledge on master planning by providing a critical evaluation of urban development strategies and offering a roadmap for future master plans.

**Keywords:** urban development; sustainable development; spatial structure; urban expansion; urban landscape; urban green infrastructure



## 1. Introduction

Most countries across the globe, particularly developing countries, are seeing a rise in the number of urban inhabitants as a result of natural population growth and the continuous movement of people from rural areas to towns and cities. According to the UN research (World Urbanization Prospects 2018), the worldwide urban population surpassed the global rural population for the first time in history in 2007. Since then, the world's urban population has grown faster than the rural population, and it is anticipated that 60% of the world's population will reside in urban regions by 2030 [1].

As cities seek to deliver high-quality services and meet rising demand, new difficulties emerge, such as rising land costs, a housing crisis, insufficient infrastructure, a lack of green space, and chaotic city expansion at the detriment of the surrounding natural environment. Consequently, cities are typically swallowing up more and more natural or rural regions [2].

The master plan stands as a cornerstone in urban governance, playing a vital role in shaping the future structure of a city.

The master plan in definition is "a dynamic long-term planning document that provides a conceptual layout to guide future growth and development. Master planning is about making the connection between buildings, social settings, and their surrounding environments. A master plan includes analysis, recommendations, and proposals for a site's population, economy, housing, transportation, community facilities, and land use.

It is based on public input, surveys, planning initiatives, existing development, physical characteristics, and social and economic conditions" [3].

Functioning as a compass for urban progress, the master plan outlines a clear trajectory for future development, tailored to meet the present and future demands of the city. However, recognizing the evolving nature of urban dynamics, the master plan should be subject to periodic review and updates by the local town planning agency. These adjustments align the plan with evolving requirements and changing circumstances, ensuring its relevance and effectiveness over time. In cases of rapid urban expansion or significant changes in urbanization patterns, the town planning agency may even require the creation of a completely new master plan every few decades, thereby accommodating the evolving cityscape and the performance evaluation of prior development approaches [4].

An essential factor underpinning the successful implementation of the master plan is the establishment of a clear and enforceable legislative framework. This framework serves as a robust foundation that enables seamless integration between planning and land administration, articulates precise planning standards, and formulates coherent land use policies [5]. The amalgamation of a well-defined legislative structure with the master plan lends authority and efficacy to urban planning and ensures the city progresses towards its envisioned future with a well-coordinated and sustainable approach.

Notably, cities worldwide have harnessed the power of master plans to elevate their infrastructure, augment public spaces, and enhance the overall quality of life for their inhabitants.

Barcelona, Montpellier, and Latakia are three cities that have notable similarities in their geographical positioning, urban expansion, and population growth. Firstly, all three cities are situated along the coast of the Mediterranean Sea, which not only fosters ecological and climatic resemblances suitable for comparative analysis, but also positions them as strategic trade and transportation hubs. Moreover, these cities have undergone substantial urban development and a surge in population over recent decades, presenting challenges linked to urbanization, thus highlighting the urgency for adopting sustainable development strategies.

Furthermore, an additional shared characteristic between Latakia and Barcelona is their roles as hosts of significant sporting events (the Mediterranean Games in 1987 in Latakia and the 1992 Olympic Games in Barcelona). These events had a pronounced impact on the urban planning strategies of both cities, leading to noteworthy developments in essential infrastructure and the creation of diverse facilities. Consequently, these endeavors played a pivotal role in driving substantial enhancements within the urban landscape, underscoring the lasting impact of these events on the cities' overall urban progress.

Barcelona and Montpellier, as European cities, have historical ties and shared urban planning practices influenced by similar cultural norms and economic frameworks. Their master plans likely draw from a heritage of urban development principles that have evolved over time within the European context. In contrast, Latakia, situated in a different geopolitical region and cultural milieu, may have developed its master plan with unique considerations specific to its local traditions, economic dynamics, and social priorities.

Despite these disparities, comparing the master plans of Barcelona, Montpellier, and Latakia remains valuable. By acknowledging the contrasting cultural and economic references, we gain a more nuanced understanding of how different cities tackle urbanization challenges and embrace sustainable development strategies based on their distinct contexts. Barcelona and Montpellier's master plans, as products of European urban planning expertise, may offer innovative solutions that have proven effective in managing urban growth, enhancing infrastructure, and improving quality of life within their European settings.

Ultimately, the comparison of these master plans serves as a valuable cross-cultural and cross-regional exploration, yielding a rich tapestry of urban planning practices. By recognizing both the similarities and differences among these plans, city planners, policymakers, and researchers can adopt a more inclusive and informed approach to urban development.

## 2. Materials and Methods

Numerous local and worldwide researchers and professionals have been researching and examining urban planning, urban expansion, and reviewing organizational plans for cities.

This study aims to analyze the effectiveness of the Latakia master plan in promoting a livable urban development within the context of international master planning practices. To achieve this, we conduct a comparative analysis of the Latakia master plan and those of two other cities with significant experience in master planning: Barcelona and Montpellier.

The research methodology includes a comprehensive analysis of the urban development strategies in the master plans of Barcelona, Montpellier, and Latakia city.

The methodology employed in this study comprises three main stages. Figure 1, shows the location and maps of the study areas for the three cities: Latakia, Montpellier, and Barcelona.

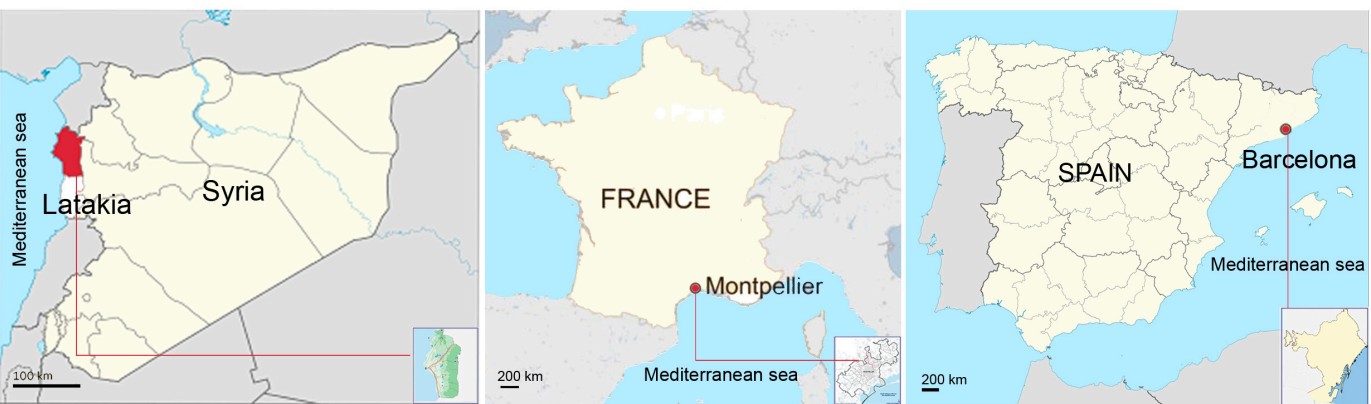

**Figure 1.** Study area locations.

A.  The initial stage entails conducting a short historical overview of the cities' growth patterns and dynamics, with the aim of gaining a deeper understanding of their urban issues.

B.  The second stage of our research involves a meticulous analysis of the urban development strategies outlined in the master plans in the three cities. We first provide an overview of the most recent master plans in all three cities. As we examine these plans, we note that the master plans in Barcelona and Montpellier undergo frequent updates and revisions, approximately every 10 years, making them renewable documents. In contrast, Latakia's master plan experiences less frequent updates, resulting in longer periods between revisions.

To ensure a comprehensive comparison, we carefully selected specific development strategies and planning documents from the latest master plans of Barcelona and Montpellier. These selections specifically focused on the first and second metropolitan strategic plans issued in 2003 and 2010 in Barcelona, along with the ScOT (Scheme of Territorial Coherence) issued in 2006. Our consideration of the dynamic and evolving nature of these strategies contributed to the robustness of our comparative process.

We then proceeded to compare these chosen development strategies with the corresponding development strategies in Latakia's master plan developed in 2008, which stands as the most recent available for Latakia. This selection serves as a fundamental reference point for our analytical evaluation, providing valuable insights into the contrasting planning approaches across the three cities.

C.  In the third stage, a comparison table will be constructed to synthesize and visually represent the findings derived from the analysis of the three master plans.

Finally, based on the outcomes of the comparison, the study concludes with the presentation of the final results and recommendations aimed at improving the Latakia

master plan, considering the identified limitations and drawing upon successful practices from the Barcelona and Montpellier master plans.

The literature background for this research is based on a wide-ranging review of relevant scholarly sources, spanning across various disciplines related to urban development. The sources on urban expansion [6–19] cover diverse topics, including the underlying causes and far-reaching consequences of urban growth. It delves into the challenges that arise in effectively managing urbanization and explores the role of urban planning as a vital tool in mitigating potential negative impacts. By examining these sources, we gain a deeper understanding of the dynamics and factors influencing urban expansion and its implications on cities and their inhabitants. The resources [20–27] address multifaceted aspects of city design, such as land use, transportation, and housing. The literature delves into the formulation of policies and strategies aimed at promoting sustainable urban development. These sources shed light on the significance of well-planned urban environments and how thoughtful design choices can create inclusive, accessible, and environmentally friendly cities. The sources [28–37] emphasize the critical role of the integration of natural elements into urban design. These elements encompass parks, gardens, green open spaces, and other forms of nature within urban landscapes. The literature highlights the manifold benefits of green infrastructure, including its potential to improve air quality, combat urban heat island effects, enhance biodiversity, and contribute to the overall well-being of urban residents, These sources advocate for the incorporation of green spaces as a means of creating healthier and more sustainable cities. Lastly, the sources on urban plan evaluation [38–41] focus on the methodologies and criteria used to assess the effectiveness of urban plans in achieving their intended goals.

These sources delve into the importance of employing both quantitative and qualitative evaluation methods. Moreover, they stress the significance of soliciting feedback from various stakeholders to gain comprehensive insights into the social, cultural, and environmental impacts of urban plans. By understanding the approaches to evaluating urban plans, decision-makers and planners can refine their strategies and ensure that future development initiatives align with the needs and aspirations of the communities they serve.

In terms of data collecting, the research is mostly based on a variety of studies that chronicle the historical growth of Latakia city and offer an evaluation of the city's spatial organization, land uses, and the effects of urban expansion on city structure [42–47]. These studies form the theoretical basis for this research. Moreover, the research collects information from official reports provided by Latakia City Council and the General Company for Engineering Studies and Technical Consultations, which contain critical information about the Master Plan 2008, the development strategies employed, and the criteria used to select the preferred alternative.

## 3. Results

Urban development strategies in the majority of developed nations have been linked to the policies and designs of balanced development that they employ at various planning levels. This has allowed them to control the effects of urban growth and curb the impact of its challenges on the planning process and the quality of life. To achieve this, master plans have been directed to accommodate this expansion by incorporating all necessary development features and elements, such as the establishment of organizational planning systems and the tools required to activate the principles of sustainable development in their communities. These development strategies also emphasize the need to address critical issues including pollution, unplanned urban sprawl, urban poverty, and other development obstacles.

### 3.1. The Case of Barcelona

Barcelona is known for being one of the leading cities in planning globally, and has addressed environmental, economic, social, and administrative issues during the past decades by working to provide large urban spaces at various levels. It has become a city

that provides an example of urban renewal planning in terms of increasing density and integrating uses while maintaining sustainable livelihoods.

Barcelona is the most densely populated city in the state of Catalonia in Spain, with a 1.6 million inhabitants. According to the 2010 census, it extends over an area of about 102.2 km$^2$, with a density of about 15 thousand people/km$^2$ [42]. It is located on the shore of the Mediterranean on the northern coast. Barcelona city belongs to the Barcelona metropolitan area (AMB), which consists of 36 municipalities with a population of 3.2 million people, according to the 2010 census.

Barcelona's modernization began with the preparations for the 1992 Olympics. Planners devised a comprehensive strategy and utilized the Games as a tool for city-wide improvements. The Olympic facilities were distributed throughout four neglected metropolitan districts. The Olympic Village, built on abandoned industrial property along the shore, was the most well-known feature of the time [48]. The most significant consequence has been the development of six artificial beaches on either side of the Olympic Port, and for the first time in its history, Barcelona faced the sea [48]. Figure 2 shows Barcelona's urban growth from 1990 until the post Olympics period.

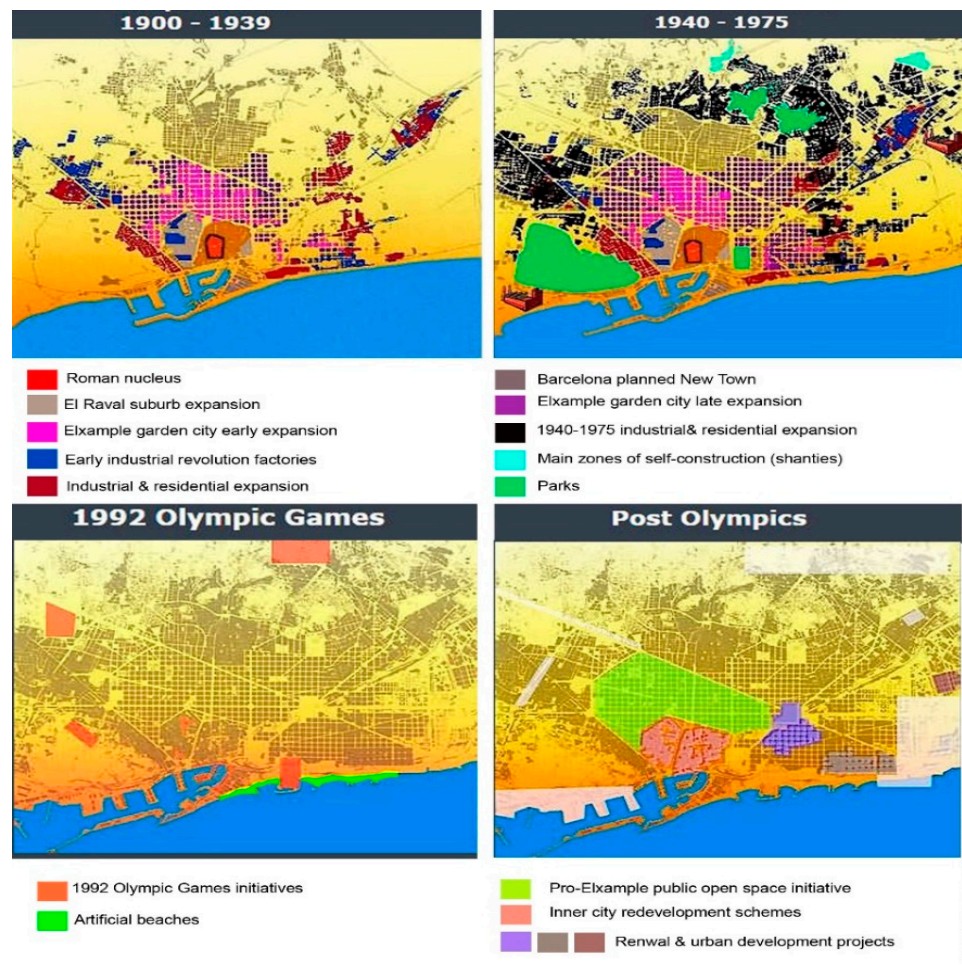

**Figure 2.** The growth and city development stages of Barcelona: Source: authors based on [48].

### 3.1.1. Beginnings of 21st Century's Urban Development Strategies

In 2003 the metropolitan strategic plan (PEMB) was approved, which came as a result of the development of the three economic and social strategic plans of Barcelona (BCN1-BCN2-BCN3) [42]. It was defined as an integrated scheme that included classifications of land uses (urban, agricultural, and land for development), and was designed with a short and long-term vision in mind, with the goal of managing the region's economic and social

development, political reform, and development as one of Europe's developed regions up to 2020.

Work continued in accordance with the principles of the first strategic plan 2003 until 2010, when the second Metropolitan Strategic Plan of Barcelona–Barcelona vision 2020) was authorized. The goals were to make the Barcelona metropolitan region a hub of innovation and creativity, attracting talent, enterprises, suitable infrastructure, and social harmony.

### 3.1.2. Barcelona Vision 2020

Barcelona Vision 2020 was approved on 2 November 2010. It was formulated by the Regional Government, the Province of Catalonia, and the Barcelona City Council in cooperation with the private sector. It is considered a comprehensive and ambitious strategic plan that has the potential to make a significant impact on the city.

The main objective of Barcelona Vision 2020 is to make Barcelona a more attractive and influential European region for innovative talent, with a quality model for social integration and cohesion.

To address these issues, the plan established five transformation levels that would serve as structural axes for overcoming the challenges [49].

-   A strong educational and university system.
-   A business-friendly administration that is quick and dependable.
-   Governance that emphasizes public–private co-responsibility in strategic project management and provides novel criteria in project management.
-   Broad language expertise that supports internationalization, talent attraction, complete integration into global markets, the Barcelona brand and an international airport.
-   Future values that complement and enhance the foundation of present and historic values, giving the city and its citizens a new personality.

Table 1 shows the main master plans in Barcelona with the main objectives for each one.

**Table 1.** The main master plans in Barcelona. Source: Author based on: [50].

| Strategy | Date/Year | Scope |
| --- | --- | --- |
| Barcelona Strategic Economic and Social Plan 2000 (BCN1) | (1990–1994) | Support the city's transformation ahead of the Olympic Games and to define a shared future vision to achieve by the year 2000. |
| 2nd Barcelona Strategic Economic and Social Plan 2000 (BCN2) | (1994–1998) | Consolidate the city's international presence once the immediate benefits had been obtained after successfully holding the Olympic Games. |
| 3rd Barcelona Strategic Economic and Social Plan (BCN3) | (1999–2005) | Position the city internationally within an incipient context of inter-city networking and included a vision of Barcelona as an open and knowledgeable city respectful of its local surroundings. |
| 1st metropolitan strategic plan | 2003 | Managing the region's economic and social development, political reform, and development as one of Europe's developed regions until 2020 |
| 2nd Metropolitan Strategic Plan of Barcelona | 2010 | To make the metropolis more attractive to attain a new economic boost. suitable infrastructure, and social harmony |

### 3.1.3. Main Urban Development Strategies in Barcelona
Housing Development Strategy

Since the middle of the twentieth century, as a result of unplanned urbanization, it has been difficult to stop low-income immigrants from building slums in unregulated lands. These agglomerations have been constructed without respect for the fundamental urban infrastructure that has emerged in the city's peripheral slopes. The inadequacy of preparations to handle the city's urban growth resulted in its expansion at the expense of agricultural fields to accept the influx since 1929 [51].

In 2009, a uniform approach, "Strategic Plan for Neighborhoods on Slope" was devised as a housing idea for unregulated peripheric residential areas [51], with a focus on social justice and adherence to the principle of mixed-use development to ensure easy access and integration with the urban network. The initial idea for the revitalization was to replace old buildings with a new eco-district, which focuses on promoting green spaces, pedestrian-friendly areas, sustainable transportation, and energy-efficient buildings. Several urban renewal initiatives were launched in the first periphery of Barcelona, which is characterized by a blend of industrial and residential areas and is inhabited by numerous working-class and immigrant communities. The promotion of social housing was a key feature of these renewal projects, as it can enhance the residents' standard of living, promote social inclusion, and stimulate economic growth.

Participation of residents was considered throughout planning and execution. A set of conditions, such as minimal energy use, water recycling systems, and the utilization of renewable energies, were agreed upon by representatives from the neighbors and the government [51]. Figure 3 shows two urban renewal projects in Peri Trinitat Nova and Peri Vivendes Del located in Nou Barris district, which are designed to address specific needs and challenges within the Nou Barris district, contributing to its overall improvement and the well-being of its residents.

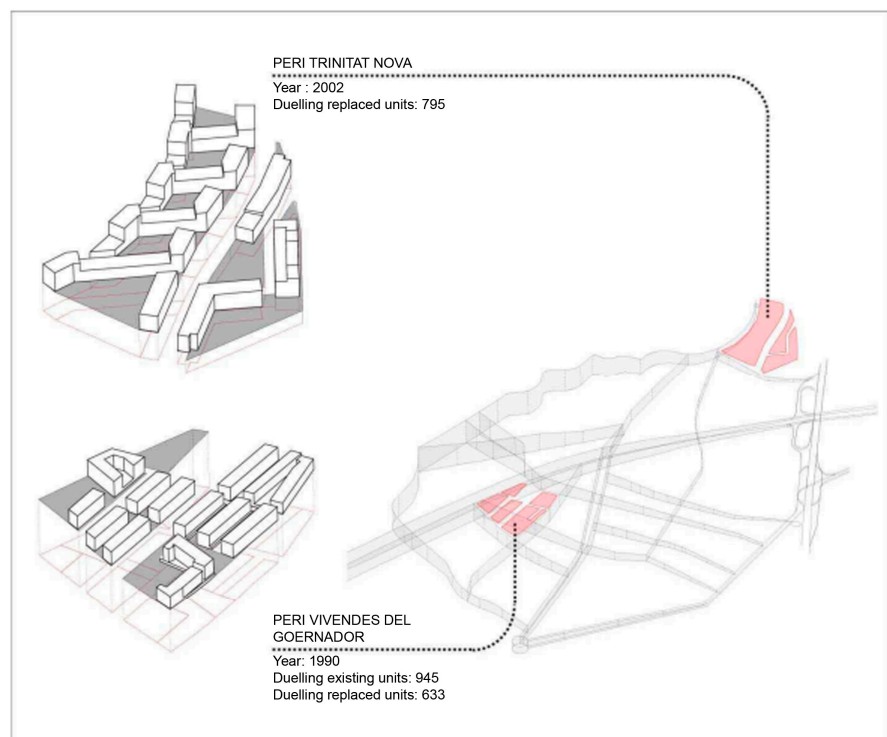

**Figure 3.** Urban renewal punctual interventions in Nou Barris district in Barcelona. Source: [51].

Furthermore, Barcelona metropolitan strategic plan has implemented various urban renewal projects that aim to improve the quality of life for residents through the enhancement of housing, public spaces, and sustainable development. For instance, the 22@ and La Segrada urban renewal projects are examples of such initiatives that have transformed old industrial areas into new urban centers, taking into consideration sustainable urban development, with a focus on improving the well-being of its citizens through the creation of high-quality public spaces and environmentally friendly features.

Urban Mobility Plan (PMU)

Barcelona has the (2013–2018) Urban Mobility Plan (PMU), a planning instrument used to outline the courses of action that will control urban mobility in the following years. It

has been designated as a strategic horizon for progress toward a more sustainable, efficient, safer, healthier, and equitable collective mobility paradigm. The plan lays out the goals and actions that must be taken to ensure that different modes of transportation and the people who use them can coexist on public highways, that pedestrians and cyclists are prioritized and protected, that public transportation is promoted, that the use of private vehicles is reduced, that commercial and tourist mobility is regulated, and that the overall efficiency of the mobility network is ensured [52].

The plan comprises the following primary courses of action:

- Organization of the city's urban pattern in superblocks and other calming measures.
- Implementation of the new orthogonal bus network and maintain the current level of traffic service.
- Total development of cycling network.
- Promotion and positive discrimination measures of high occupancy vehicles.
- Compliance with regulatory environmental quality parameters
- Revision of parking regulations on- and off-road.
- Improving loading and unloading efficiency.

Following the implementation of the mobility urban plan, Barcelona has seen extraordinary achievements, including a 67 percent rise in cycling, a 21 percent drop in the use of private automobiles, and a 3.5 percent increase in the use of public transportation [52].

The "superblock" (superilla in Catalan) is a concept that originated in Barcelona, and involves grouping several conventional city blocks together and reconfiguring the streets within them to create a large, pedestrian-friendly block with reduced traffic and more green space.

Under the superblock plan, streets within the block are narrowed, speed limits are lowered, and priority is given to pedestrians and cyclists. Cars are allowed to enter the block only to access parking, and through traffic is rerouted around the outside of the superblock. This creates a calmer, more livable environment with less noise and air pollution, and encourages active transportation, such as walking and cycling. Figure 4 shows the super block function plan and transportation mode impact.

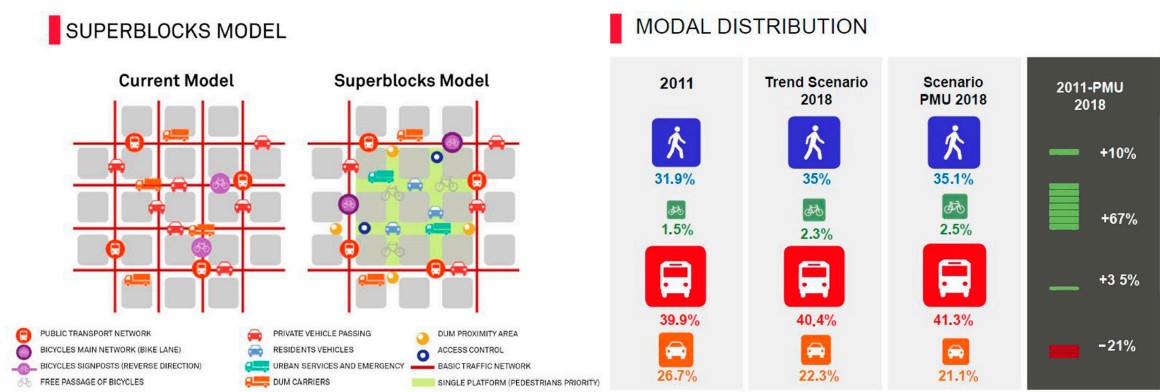

**Figure 4.** Superblocks model and modal distribution of (PMU) in Barcelona. Source: [52].

Sustainable Planning for the Natural Areas and Biodiversity of Barcelona

Barcelona's policy of protecting natural areas and green spaces evolved from the compressed city policy to provide green and public spaces within the city. It adopted the policy of continuously producing urban public sites [53], based on the self-sustaining urban agricultural ecosystem that was implemented in Barcelona to address the problems it faced due to its shrinkage beginning in the 1980s, its population orientation towards the city periphery, and the emergence of a new urban agricultural ecosystem.

Urban agriculture policies, which have been considered since the first metropolitan master plan, have been instrumental in improving the availability of parks, particularly in residential areas. The strategy (sustainable tourism, leisure, employment, and trade)

supports the idea of creating development corridors for the population, whether in terms of providing sustainable transportation that minimizes negative environmental impacts or of opportunities for sustainable urban development that considers the social, economic, and environmental impacts of urban growth; such opportunities might include the establishment of an urban farming system and pedestrian and bicycle paths, and the creation of an interconnection between green areas, urban agriculture, and residential areas. Figure 5 shows urban agriculture system proposed in the strategic planning for the Tres Turons hills

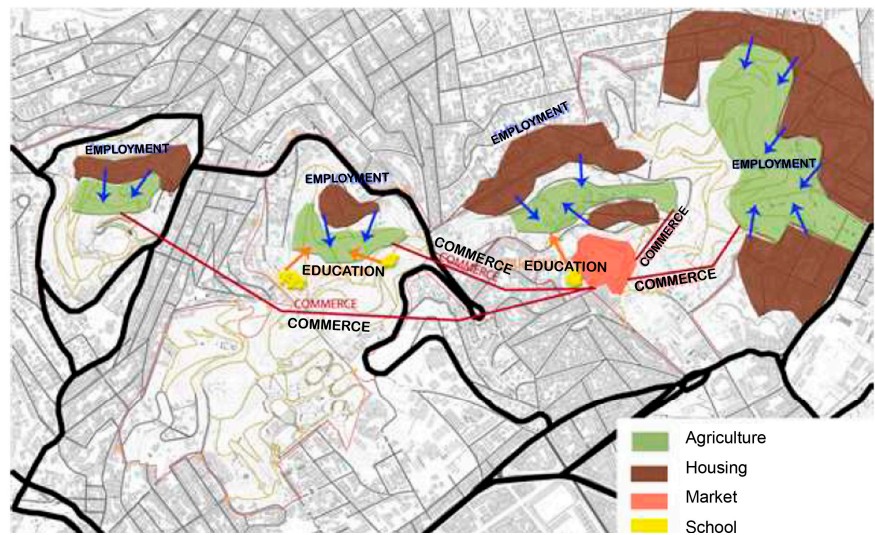

**Figure 5.** Urban agriculture system in Tres Turons hills in Barcelona. Source: [53].

Thérèse-Toros Park, which is located in the northern part of Barcelona, is considered a significant example of the use of urban agriculture strategy. The park incorporates a communication network of corridors, nodes, and loops, as well as sustainable transport and an urban farming system, all designed to promote ecological and social sustainability.

The inclusion of urban agriculture within the park's design is particularly significant, as it not only provides a source of fresh produce for local residents but also contributes to the park's overall ecological sustainability by promoting biodiversity and connecting different natural areas within the park. Additionally, the park's multifunctional design, which includes recreational areas, spaces for urban agriculture, and interconnections between green areas and residential areas, creates opportunities for social interaction and community engagement while also promoting physical activity and healthy living. Overall, the Thérèse-Toros Park serves as an example of how the integration of urban agriculture into a larger master plan for sustainable development can enhance green spaces and promote both ecological and social sustainability in urban areas.

### 3.2. The Case of Montpellier

Montpellier is the capital of the Hérault department and the second largest city in the of southern France, with a city center 7 m (12 km) from the Mediterranean coast. Montpellier is the Occitanie region's administrative and commercial capital. The city location is on a fertile plain, and the city has built up around its historic districts, which are now surrounded by boulevards on the original city wall locations. It is known for the Promenade du Peyrou, a terraced 17th- and 18th-century promenade with views of the Mediterranean Sea.

Montpellier was a grape center in the early twentieth century with little urban transformation and population expansion. During this time, Montpellier's service sector (universities, high schools) was strengthened, and some new infrastructure was developed (monumental places, theatre, museums, etc.). Following World War I, the first low-cost housing was built, as well as some public services (more schools, university equipment, a

big hospital). Simultaneously, army territory was converted to civil usage, resulting in an increase in built-up areas [54].

In 1956, Montpellier was upgraded to become the capital of Languedoc-Roussillon, and the 1960s were years of demographic, economic, and spatial growth: public and private investments, new highways, and new periphery districts. Montpellier grew from 120,000 people in 1962 to 200,000 people in 1982 [54]. Montpellier was opened up to a massive economic field during this time, with the establishment of "new economy" actors (such as IBM in 1965), tourism infrastructure in the area (new seaside sites), and transportation equipment. With more economic expansion, social evolution, and urban development initiatives, the Montpellier area became a site defined by a young population (students, young people working in new industries), a large demographic boom, and a high demand for housing, which was met by expanding the urban area. As of 1999, the urban area of Montpellier accommodated a population of 460,000 residents [54].

During the 35 years from 1968 to 2004, more than 15,000 hectares have been used, approximately half of this area corresponding to residential occupation and the rest to the other diverse occupations of space to which urban practices lead: (infrastructure), amenities and leisure facilities, commercial activities, education and culture (community facilities), and work (activity areas). It is considered that each additional inhabitant has led, in the past, to the additional use of approximately 800 m$^2$: 400 m$^2$ for residential use and 400 m$^2$ for other uses [55]. Figure 6 shows the urban expansion in Montpellier between 1952–1999.

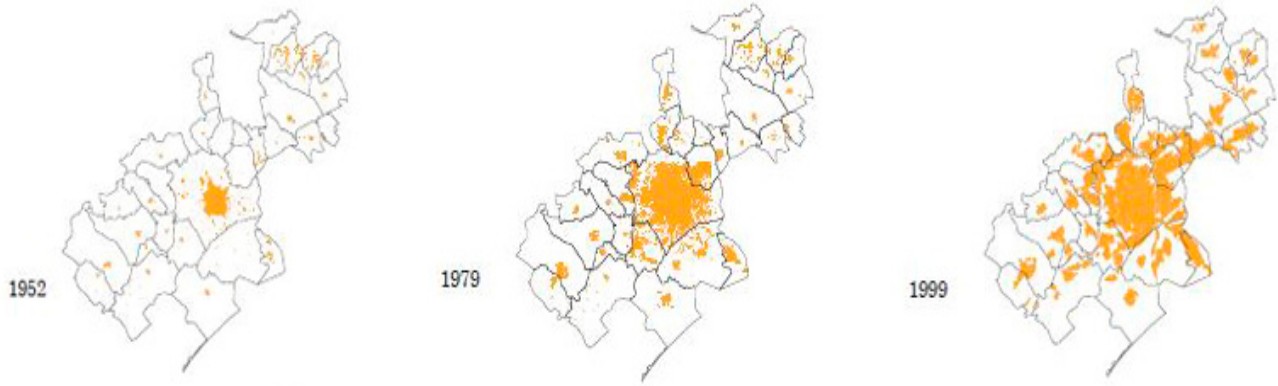

**Figure 6.** Urban expansion in Montpellier between 1952–1999. Source: [55].

3.2.1. City Development Strategies

Montpellier's SCOT (Scheme of Territorial Coherence) is a planning tool used for urban and rural contexts, including the urban fringe. The SCOT imposes spatial planning at the inter-communal level with the scheme of territorial coherence and sets the main planning orientations in the Montpellier Agglomeration for the next 10 years [54].

The basic idea of the Montpellier city plan SCOT (which was approved by the Community council in February 2006) as endorsed by the report on the "Local Plan for Urban Development of Montpellier (PLU)" and the report on the Project Management and Sustainable Development (PADD), is based on the integration of three main axes (environment-, economic-, and urban development) that ultimately aim at integration in the development of the Montpellier region, and at the same time the development of sustainability for all sectors [56].

Economic Development

The economic development strategy of the SCOT scheme in the Montpellier master plan aims to promote a diversified and resilient local economy that is less dependent on any one industry or sector. This involves supporting small and medium-sized enterprises, as well as promoting innovation and entrepreneurship. Additionally, the scheme recognizes the significance of tourism to the local economy and strives to encourage sustainable tourism practices that respect the region's natural and cultural heritage, including eco-

tourism and cultural tourism. To further facilitate economic growth, the SCOT scheme also emphasizes the importance of investing in infrastructure and public services, such as transportation infrastructure, education, healthcare, and public transit. Furthermore, the scheme acknowledges that effective economic development requires collaboration and partnership between government, business, and civil society stakeholders. Thus, the scheme aims to foster partnerships and collaboration to promote sustainable economic growth and development in the region.

Urban Development

The urban development strategy employed by the SCOT scheme in the Montpellier master plan aims to create a sustainable, livable, and connected urban environment by:

- Reducing urban sprawl and finding solutions to accommodate the expected population growth of the city, such as available vacant lands and lands suitable for development and encouraging growth in low-density areas.
- Establishment of socially diverse housing projects (respecting the principle of social mixing and avoiding segregation areas for medium and popular housing).
- Working with the principle of mixed uses, which means maintaining a diversity of uses in the same area between commercial and economic activities and housing.
- Strengthening the communication between the neighborhoods of the same city, as well as between the cities of the region, through the use of highways and paved roads, as well as of internal tram lines, which contribute significantly to strengthening communication between parts of the entire region, as well as to the ease of access from any part of the region, from the province to the beach areas.
- Reducing the use of cars within the city and providing means of transport that are easy and non-polluting for the environment, such as cycling paths.

Figure 7 shows the tram lines service range and the suggested development directions within the city, and shows the hotspots, which refer to areas that have been identified as having high potential for development. The SCOT scheme prioritizes investment in these areas to stimulate economic growth, improve public spaces and amenities, and promote sustainable urban development.

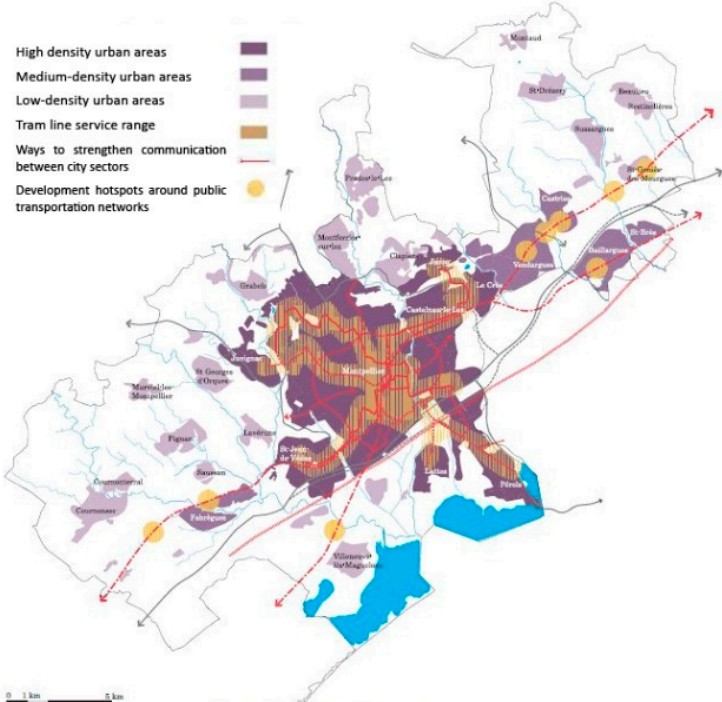

**Figure 7.** The dynamics of urban development around the public transport network in Montpellier. Source: [57].

Sustainable Environmental Development

The SCOT scheme in Montpellier master plan prioritizes the preservation and enhancement of natural resources, such as wetlands, rivers, and forests, and aims to establish a green network that interconnects these natural areas to promote wildlife movement and ecological resilience. The green network is also designed to facilitate recreational activities, including hiking, cycling, and birdwatching, thereby promoting physical activity and improving residents' quality of life. Furthermore, the green network serves as a natural filter that can absorb pollutants and mitigate the negative impacts of urbanization on air and water quality in Montpellier. Figure 8 shows the employed green network techniques within Montpellier's environmental development strategy.

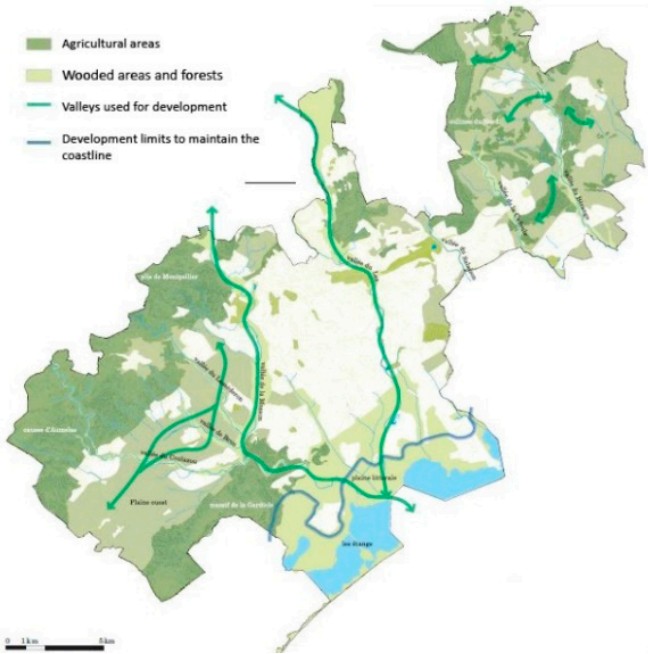

**Figure 8.** Environmental development of the Montpellier agglomeration. Source: [54].

Housing Policy

Montpellier's housing policy aims to meet the different housing needs while respecting the principle of social and urban diversity to avoid any segregation in the urban space. The Local Housing Program (P.L.H) is the local application of the SCoT in terms of housing. It allows the detailing of SCoT objectives on a local scale, and it is part of the strategy developed at the level of the Agglomeration, which is formalized in the inter-municipal agreement.

In this area, the policy of the City of Montpellier is based on three main orientations:

- diversify the production of housing in new urban areas;
- act on the housing stock of existing districts;
- respond to categorical housing needs that are not satisfied by the mechanisms of the market.

In order to achieve diversified housing production in future urbanization areas, the city relies on the approach of diversifying housing production to meet all needs by avoiding excessive residential specialization (residential complexes) and allowing all neighborhoods to provide the greatest diversity in housing, especially social rental housing, in addition to supporting and a preference for building collective housing to reduce urban sprawl.

The urban development plan of the city passed a law stipulating that 25% to 30% of housing built in new neighborhoods must be social rental housing. The city's social housing stock was estimated at 28,858 in 2012, which represented 21.74% of the number of main housing units [56].

However, the rapid growth of the population and the high demand for housing make it necessary to maintain this trend. Therefore, the municipality of Montpellier has included in its PLU (Plan Local d'Urbanisme) a provision that makes it mandatory in all sectors for residential use to build social housing subsidized by the state. This provision applies to any project with an area of more than 1200 square meters, with the exception of tourist housing, and gradually according to the size of the project [56]. Figure 9, shows the planned and executed Proportion of social housing within the general planning process in Montpellier

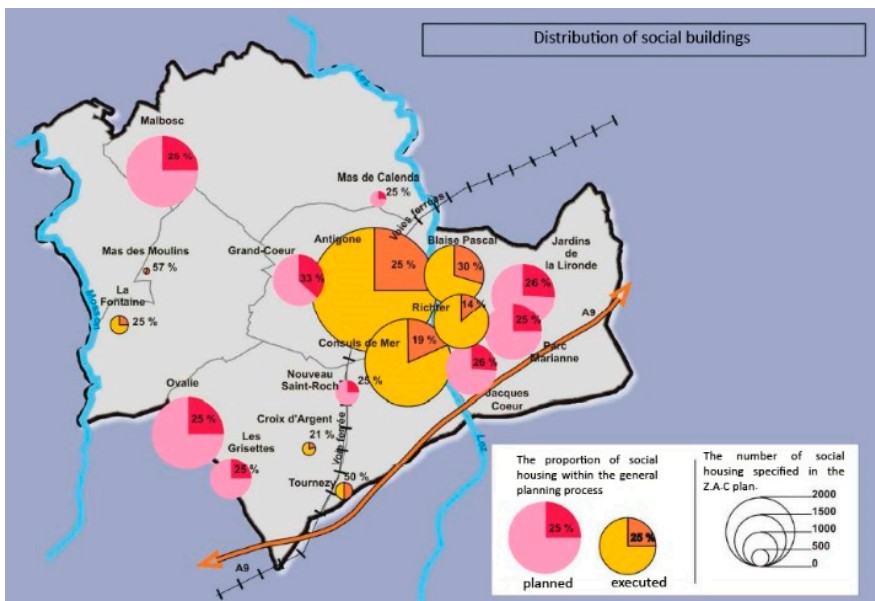

**Figure 9.** Proportion of social housing within the general planning process in Montpellier. Source: [56].

Urban Mobility

The city's objective is to mitigate the use of private vehicles between the outskirts and the city center by promoting public transportation and non-motorized modes of travel, particularly cycling, as potential alternatives to cars. The development of an efficient public transportation system is deemed necessary to manage the growing volume of commuter traffic between residential and work areas. The adoption of a tram system as a solution allows for the reorganization of urban spaces, as it is accompanied by the revitalization of public areas along its route. Therefore, the tram system is considered a viable alternative to private cars for urban mobility [56]. With regard to motorized travel, four orientations have been presented in the PDU (Plan of Urban Mobility), which is a planning document considered as the specific part of SCoT addressing mobility and transportation [54], and the four orientations are:

- develop the public transport network;
- organize car traffic on a hierarchical network of roads favoring bypasses, belts, and connections between districts that do not require passing through the city center;
- continue the parking policy by favoring free parking for residents near the city center and by creating relay parks on the outskirts to access the tramway;
- encourage companies and public authorities to draw up a mobility plan for their staff (the Company Travel Plans).

To reduce the impact of transportation in the city and maintain the landscape connection within the city, priority is given to soft circulations that prioritizes non-motorized modes of transportation, such as walking and cycling, at both the local and agglomeration levels. These circulations provide a genuine connection between the various landscape typologies and keep green paths going continuously through the city areas [54]. Figure 10, shows the soft circulation strategy in Montpellier.

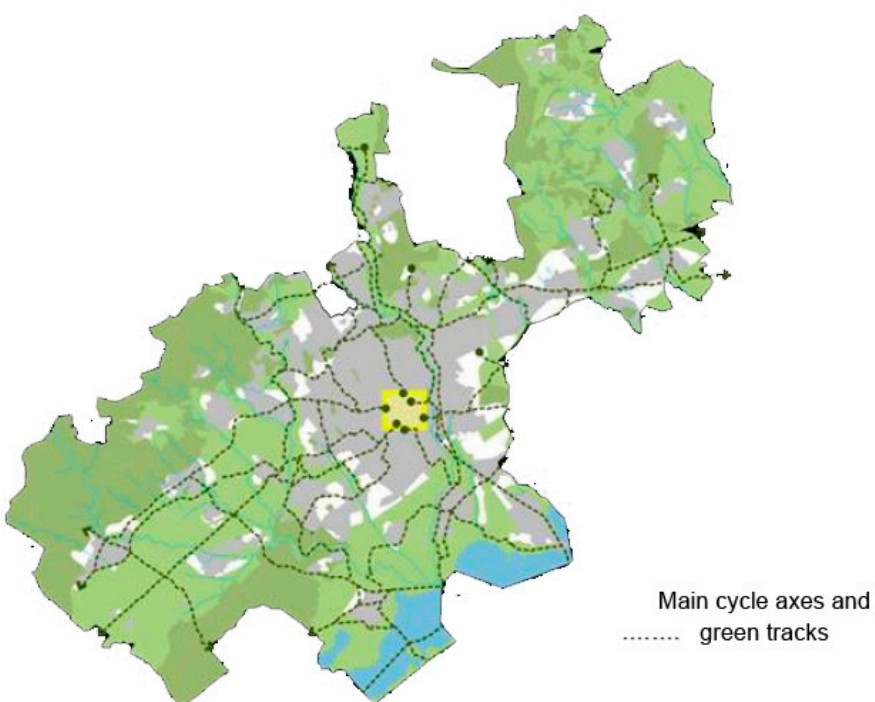

**Figure 10.** Continuous landscape (soft circulation method) Montpellier agglomeration. Source: [54].

*3.3. The Case of Latakia*

The Syrian cities suffer from major urban problems as a result of the inadequacy of the organizational plans and the inefficiency of the master plan used in developing the city and determining the directions of its future growth. This is because these plans did not draw on clear planning standards, but rather were formulated according to standards for organizing land use determined by the legislative decree No. /5/ for the year 1982 [43]. Which are traditional quantitative standards for building density, population and transportation.

Latakia governate is located on the northern part of the Syrian coast with an area of 244,000 hectares, which constitutes 1.2% of the total area of Syria [42].

Latakia governorate is divided into four administrative districts (Latakia-Jableh-Al Qurdaha-Al Haffah), and each of these administrative districts has a number of sub-districts, villages and towns. Latakia district, positioned as the central and principal administrative hub within the governorate, commands a substantial demographic influence, constituting 51% of the entire governorate population. With a district populace of 603,831, it assumes a pivotal role in relation to the overall governorate population of 1,080,000 [42]. Figure 11 shows the administrative division for Latakia governorate.

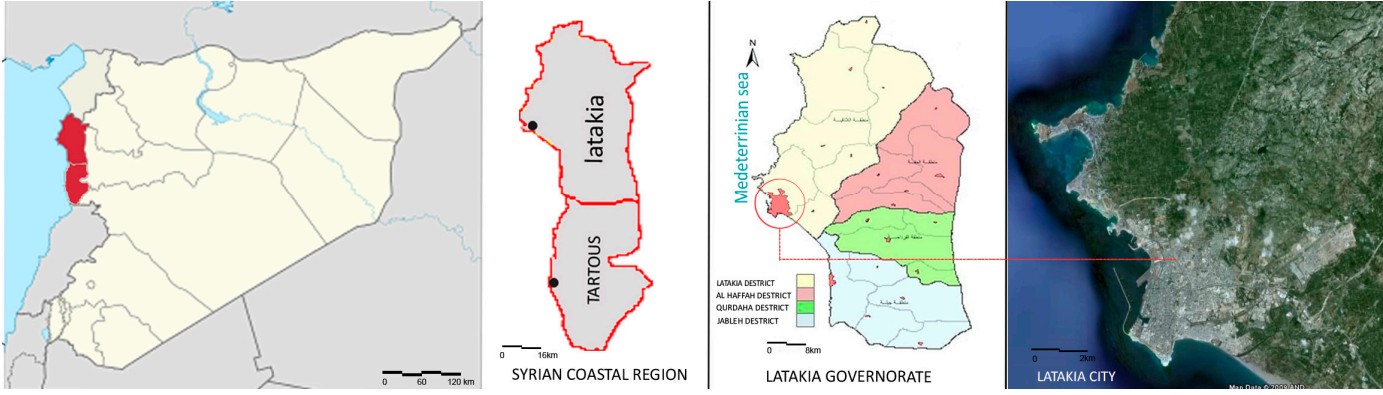

**Figure 11.** Latakia governorate administrative division and city map.

### 3.3.1. Latakia City Urban Structure [44]

Latakia district is divided into 20 neighborhoods. Each has a distinctive population, density, and area. The highest population density is in the western neighborhoods of the city, which are associated with both commercial and residential activities.

Area 1 (11 neighborhoods): This area represents the modern part of the city, which took the place of the Old City during the 1980s and 1990s period of significant urban transformation. The majority of the city's marketplaces and logistical infrastructure are located in this highly populated neighborhood.

Area 2 (5 neighborhoods): This area has business, residential, and agricultural activities. Low-density informal housing was created when the rich agricultural fields were progressively urbanized.

Area 3 (4 secondary settlements): Previously held by tiny secondary communities (initial population did not exceed 13,000 in 2010 estimates), which developed in Latakia's perimeter and attracted urban growth. Recently this area has been incorporated into the city municipal authority.

Together, Areas 1 and 2 make up the majority of Latakia's urban area (8300 ha). This area also has significant logistical transportation facilities, including a port, two duty-free zones, and complex rail and road systems.

Latakia city is recognized as one of the cities experiencing fast urban expansion. It has expanded along with other cities as a result of both organized and disorganized urban sprawl, and its built-up areas have evolved in a variety of ways to meet the growth that is occurring there. These changes undoubtedly contributed to the city's development and brought about adjustments that led to demands and issues that must be considered while creating new urban development plans. Figure 12 shows the urban composition and the population density in latakia administrative district.

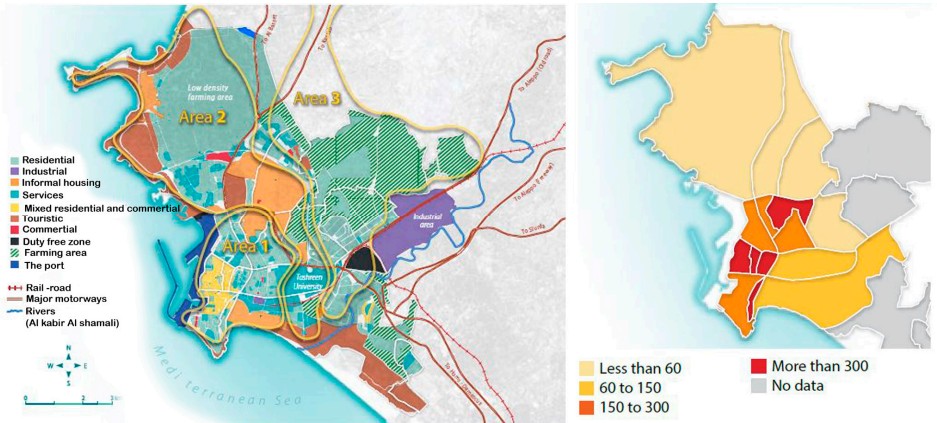

**Figure 12.** Latakia city urban composition & population density (people/ha). Source: [44].

### 3.3.2. Latakia Urban Master Plans

The first master plan for the city was prepared in 1951; the area of the city at that time was estimated at 200 hectares [42]. In 1964, the plan was developed to accommodate 250,000 people. The municipality issued laws for the expansion and extension of the city and set administrative boundaries for it. In 1976, a new master plan was drawn up for the city by the Arab Consulting Engineers Group on the basis of analytical studies for the future of the city, as the inhabited residential area at that time included 217,533 people with a built-up area of 656 hectares [45]. The plan also estimated the possibility of doubling the city's population and the built-up area until the year 2000, and solutions were developed for potential problems in increasing the number of automated means of transportation in the neighborhoods of the modern city and large villages, which were expected to become suburbs attached to the spatial structure of the city. In addition, this plan took into account

the topographical situation of the city (which was missing from the previous plan) and the general economic situation (agricultural, industrial, tourism), especially land uses.

In 1984, the new master plan for the city was ratified with an area of 3190 hectares. The new master plan focused on finding areas for the future expansion of the city to accommodate 500,000 people. Organizational plans were drawn up for the future residential communities surrounding the city. After that, the city began to spread to the north and east until it extended over an area of 4348 hectares in 1991 [42]. Figure 13 shows the growth of the urban area for latakia district.

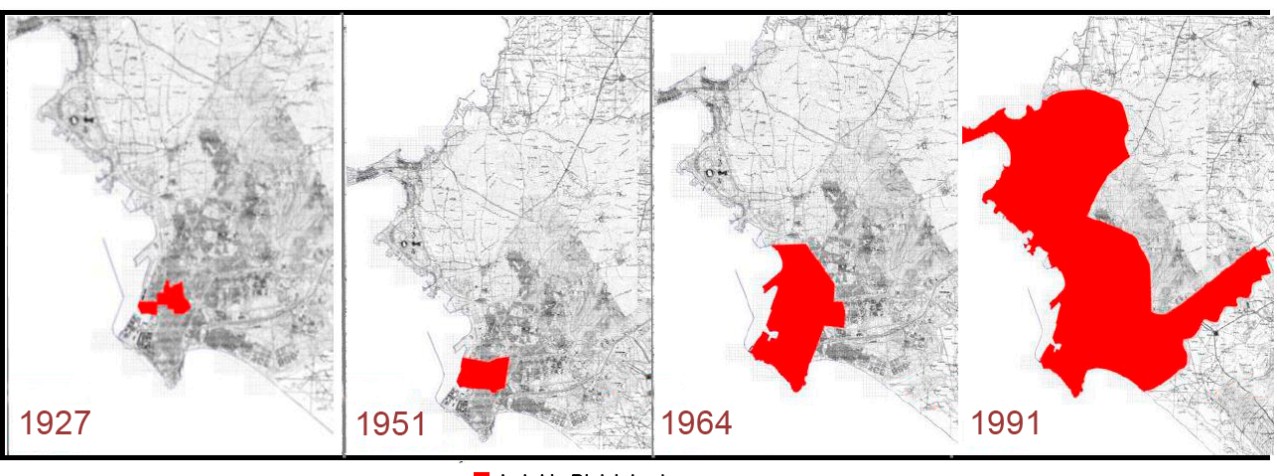

■ **Latakia District urban area**

**Figure 13.** Latakia city urban growth. Source: [42].

Despite the fact that the new master plan attempted to outline the areas where the city would expand in the future, slums began to appear and spread haphazardly between the city limits and planned residential neighborhoods. That was an inevitable result of the plan's inability to absorb the growing demand for housing with the increase in immigration from rural areas to the city, which was due to the lack of services in rural areas and the concentration of most of them in the city. These slums spread over an area estimated at 21% of the area of the full scheme, and their combined population was estimated at 180,000 people [42]. The expansion came at the expense of open spaces, green areas, and agricultural lands surrounding the city, which caused an increase in population density and a decline in the quality of life in these settlements.

The 1984 master plan was based on a study of the city's growth absorption by defining areas for expansion in the general plan on three assumptions for the expected population increase. The strongest theory was that the growth factor would be 48.1 per thousand for the period between 1975–1990 and 44.1 per thousand for the period 1990–2000.

However, due to the city's declining yearly population growth factor, the predicted density was higher than the reality, particularly in the second stage [42]. Additionally, there were differences in density between the earlier phases, which is a natural phenomenon often imposed by urban expansion. The proposed expansion areas served as the foundation for the 1984 master plan, and it should be highlighted that this strategy left a significant gap between securing the services of these expansion areas and organized residential communities located on the outskirts of the city, thus increasing the consumption of agricultural land, road lengths, and travel time.

In 1987, Latakia's hosting of the Mediterranean Games marked Syria's debut in organizing major international sporting events involving 18 countries. The occasion spurred profound urban development, creating a sprawling 156-hectare sports city, including sports facilities with green public spaces. This versatile complex not only became a hub for athletic training and events but also evolved into a vibrant center for festivals and entertainment activities, enriching Latakia's cultural landscape. The Games spurred the construction of

stadiums, training centers, and improved infrastructure, enhancing the city's overall urban planning and public services.

In 2001, the General Company for Engineering Studies and Technical Consultations obtained permission from the city council to study and issue a new master plan for Latakia city. The study was conducted in three phases with a duration of 36 months. The project was supposed to be completed in the year 2004, but the announcement of the master plan was postponed several times in order to keep pace with the urban development of the city; it was announced in 2008. The area of the city in the new plan amounted to 10,034 hectares, with an area of 4800 hectares more than the plan announced in 1984 [57]. Figure 14 shows the city boundaries within latakia master plans 1984 and 2008.

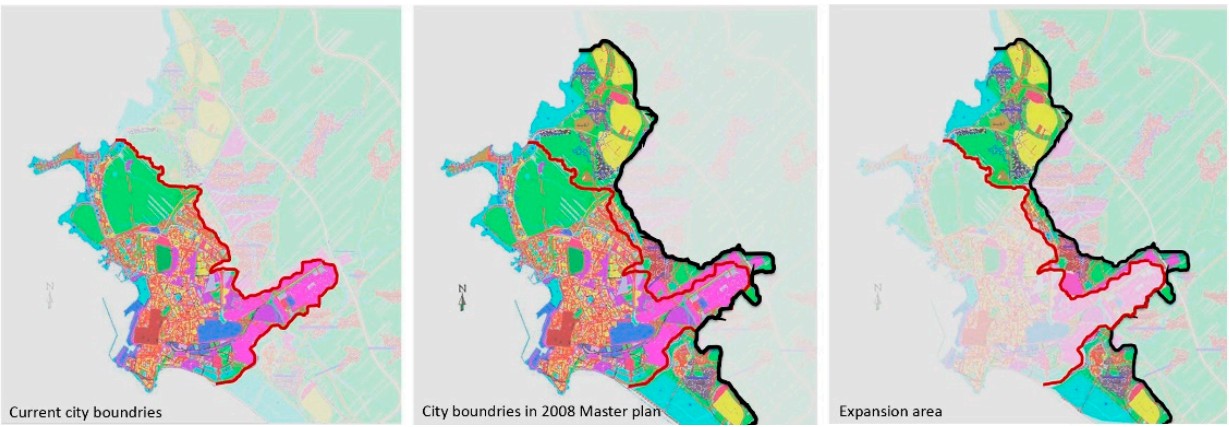

**Figure 14.** Latakia City boundaries regarding the 2008 master plan. Source: [57].

In order to develop the final master plan for the city, five alternatives were proposed for urban development of the city. These alternatives were studied and evaluated according to several planning and design criteria.

Table 2 shows the five alternatives of the 2008 master plan after the analysis phase as it was presented by the General Company for Engineering Studies and Technical Consultations in 2004.

**Table 2.** Alternatives of the 2008 master plan of Latakia city. Source: [58].

| Alternative | Objectives |
| --- | --- |
| 1 | Continuation of the previous general master plan approach, with city expansion, organization of agricultural areas, and road network adjustments |
| 2 | Defining the city boundaries with a highway, utilizing available land for housing, developing the Marj area, and ensuring good connectivity to public networks. |
| 3 | Organizing part of the agricultural area in Dumsarkho, improving surrounding main roads, and developing nearby population clusters. |
| 4 | Achieving a greater balance between the city and nearby population centers, utilizing them to create new suburbs and developing tourism along the coast. |
| 5 | Linear extension of the city from the northern and southern beaches towards Jableh, developing residential areas, and connecting them with a highway. |

Finally, the approved alternative was a result of combining Alternatives 4 and 5, which achieved the highest percentage of conformity to the proposed planning standards. The vision of the plan was introduced in main points [58]:

1. The adoption of the strip expansion of the city achieves the appropriate aesthetic goals for placing the city on the sea.
2. Population and tourism development is achieved with the same strategic importance for the future.

3.  It achieves the strategic and future regional goals in terms of creating a demographic balance between the city and the surrounding and nearby population centers, thus reducing the burden on the city and promoting new urban centers.
4.  Achieving these new urban gatherings around the city and in its territory in the future and in the long run works and contributes to curbing the flow of immigration to the city.
5.  This alternative achieves long-term future orientations towards the integrated development of regional tourism, north and south.
6.  The proposal of maritime passenger transport routes contributes to the revitalization of tourism development and contributes to alleviating the burden on public roads. Figure 15 shows the main development strategies in Latakia Master plan 2008.

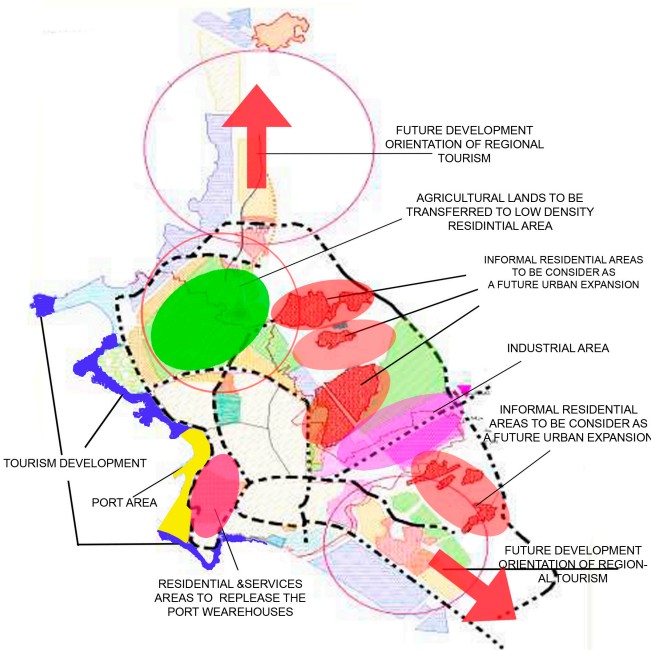

**Figure 15.** Development strategies in Latakia Master Plan 2008. Source: author based on [43].

3.3.3. Development Strategies in the Master Plan

Urban Expansion Strategy

In the population studies of the city, the population of the city was estimated at 831,000 people in 2025, according to the adoption of a population growth factor of 3 per 1000, divided as follows: 560,000 residents within the current spatial area of the city, and 270,000 residents in the expansion areas [59]. In order to accommodate this population increase, two expansion approaches were presented:

The first approach was represented by external expansion outside the current urban structure limits. In this regard, eight zones of urban expansion were defined. The area of these expansions was 1226 hectares. Five expansion areas were proposed to accommodate the population growth of the city during the first ten years, which was expected to reach 138,000 residents. These five areas were estimated to accommodate 96,000 residents [45]. Figure 16 shows the proposed zones for future urban expansions in Latakia 2008 Master plan.

The second approach represented internal expansion, by filling the gaps, accommodating the empty spaces, and utilizing the potential for absorption of the irregular areas after including these areas within the limits of the new plan and reordering them. Several informal residential areas appeared in the city, with an area of 655 hectares, or 21% of the city's built-up area. The population of these areas reached 187,000 people, with an average density of 390 people/hectare. Since these areas occupy important sites within the city

boundaries and with connections to the organized neighborhoods, it was necessary to develop a plan to reorganize these areas, link them with the organized plan, and raise the quality of life in them. The plan included:

- Connecting these areas with each other on the one hand and with the neighborhood and the city in major ways on the other hand
- Determine zones for installation, maintenance, and restoration of the services in these areas.
- Determine zones of urban developing intervention through:
- Taking advantage of the vacant lands that were not subject to the organized plan and proposing a new urban organization that allowed the transfer of residents to these areas after the completion of their construction.
- Securing the necessary services (educational, health, commercial) according to the possibilities available from vacant lands.
- Improving existing road sections as much as possible and securing connecting roads with the main transportation network.

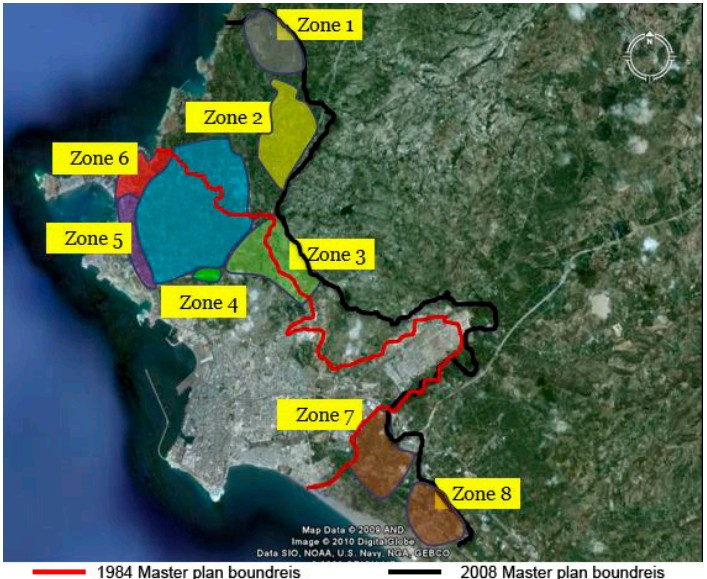

**Figure 16.** Proposed expansion zones in Latakia Master Plan 2008. Source: [57].

Figure 17 shows the informal residential areas proposed to be developed in Latakia Master plan 2008.

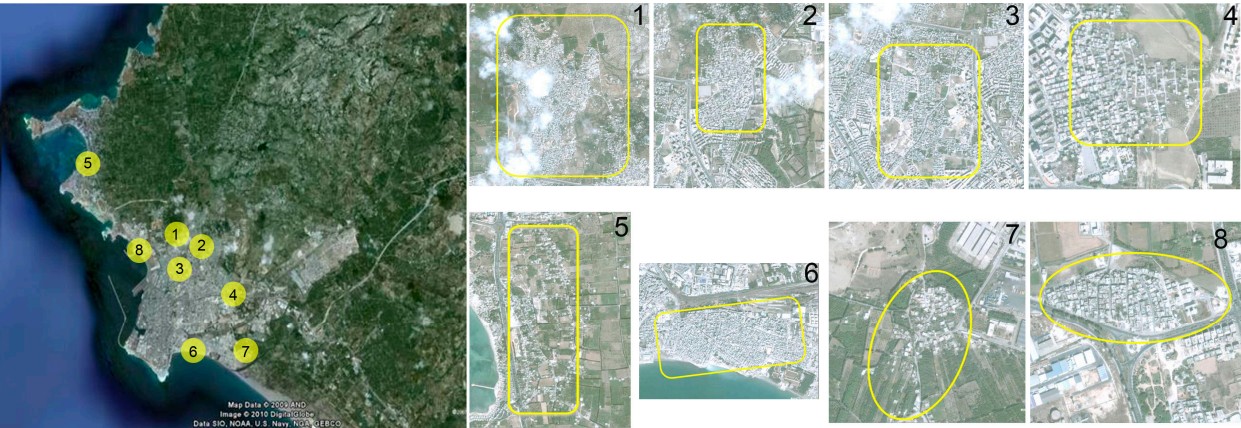

**Figure 17.** Informal residential areas to be organized in Latakia Master Plan 2008. Source: [57].

The proposed expansion strategy provided good suggestions for organizing random housing areas and determining areas for future growth, but these enhancements were only general suggestions that skipped over crucial information about the social structure of these areas, which are thought to be closer to the rural system and require an intervention mechanism that ensures maintaining the structure of these communities and integrating them into the urban framework. Furthermore, neither the environmental solutions that might be employed to meet the future needs of these places' infrastructure nor the process for developing it were discussed in this study, which might cause serious challenges in the future.

Housing Development Strategy

The housing shortage is regarded as one of the most serious issues facing Latakia governorate in general, and the city center in particular. Planners tried to provide some solutions to this problem in the new master plan. Therefore, in addition to structuring informal areas and improving their living conditions, the housing strategy aimed to achieve a better balance between urban districts and the nearby semi-urban areas by including these areas in the newly structured plan and dispersing future population growth there.

The housing development strategy proposed:

- Involving some of the suburban settlements outside the city in the new planned strategy and investing in bare land by starting new housing projects.
- Putting the slums in order, enclosing them, and connecting them to the organized areas.
- That urban expansion would likely go in the direction of the north and north-east for housing, with a chance that it will also move south as a result of the port warehouses' transformation into an urban region.

Here, we find that housing strategy relied on proposing new residential areas in the city without going deeper to find solutions to the current problems that this sector suffers from. These include a duplication problem with housing, as at the same time as housing demand is increasing, there are many vacant housing units, amounting to 20% of the governorate's housing units, which is 286,869 houses for the year 2010, including 140,245 houses in urban centers [42]. These vacant houses are mostly in the city center and belong to a small percentage of the owners, and their prices are very high, exceeding the purchasing power of those with limited income. This drives many families to settle in irregular areas despite poor services and poor conditions.

Urban Mobility and Public Transportation

Due to the concentration of the governorate's main government centers in Latakia city and the consequent increase in daily movement to and from the city, as well as the rise in the number of private cars, which worsens traffic congestion and makes moving around the city difficult, the demand for transportation in the city has increased significantly, placing a heavy burden on the public transportation system.

The road network is separated into main streets with heavy traffic connecting the center with the main sections of the city, and secondary streets that cross them and go into residential neighborhoods. Secondary streets make up 69% of the overall road network's area, while main streets make up 31% of it [60]. Figure 18 shows the correlation between the public transportation and the road network in latakia city.

A total of 101 buses run on the nine lines that make up Latakia's public transportation system. Altogether, 82% of the public transportation network is physically coherent with the major thoroughfares that serve as hubs for the everyday flow of people inside the city. While 18% of the public transport network agrees with secondary roads, which serve urban areas within the city [60].

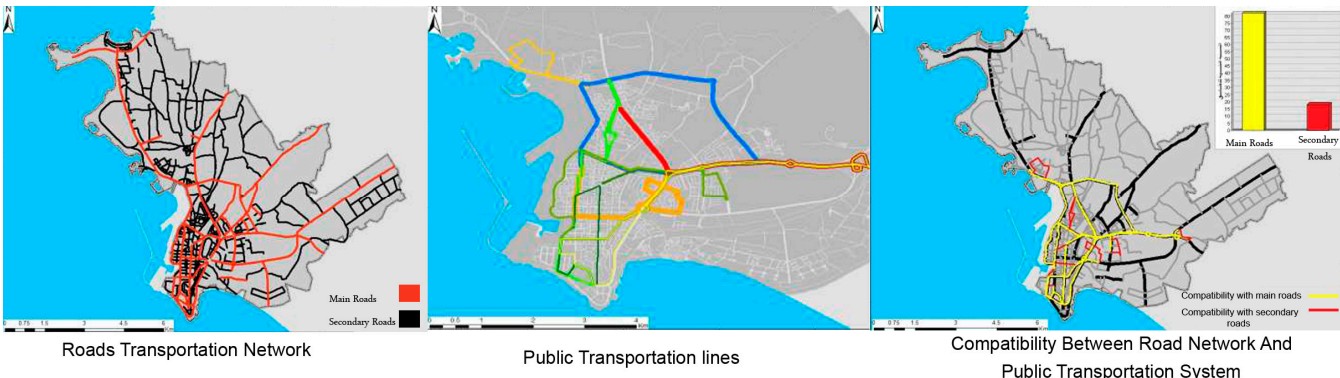

| | |
| --- | --- |
| Roads Transportation Network | Public Transportation lines |
| | Compatibility Between Road Network And Public Transportation System |

**Figure 18.** Public transportation network in Latakia city. Source: [60].

In order to relieve congestion in the city center, it was suggested to build a ring road to link the northern and southern urban areas, as well as a number of roads to connect the new expansion areas. However, no public transportation system was suggested; instead, plenty of parking spaces were allocated, which in turn encouraged the use of private vehicles and had a negative impact on the city's traffic.

The master plan proposed to combine central and decentralized roads for the distribution of services, and this merger caused confusion between local and regional traffic, reduced the safety factor in the vicinity of the residential environment, and weakened social communication [42]. Pedestrian paths and roads of their own were also proposed, but at specific hours. In order to promote tourism, the plan suggested that maritime transportation lines should be established connecting the coastal tourist spots along the shoreline. Figure 19 shows the road network development proposed in Latakia Master plan 2008.

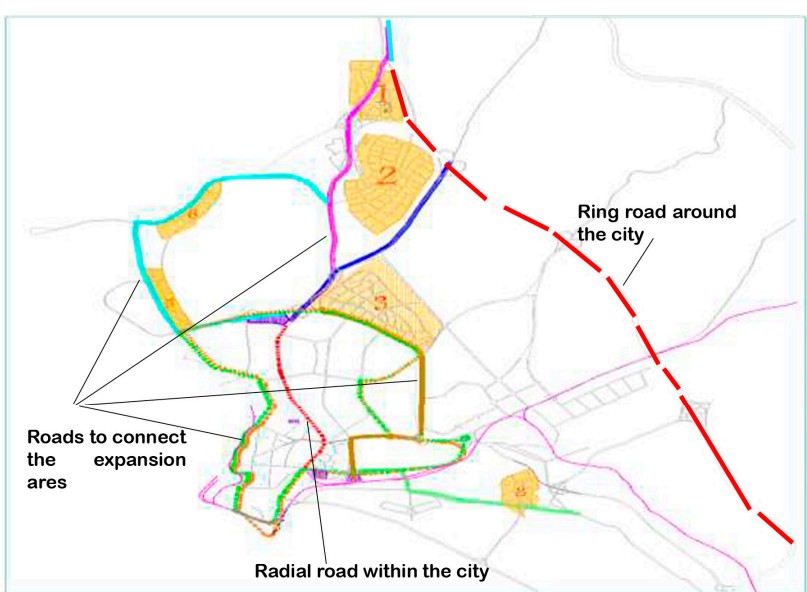

**Figure 19.** Road network proposals in Latakia master plan 2008. Source: author based on [43].

The plan did not include a mechanism to improve road sections in the city and improve pedestrian movement and cycling lines, nor did it include a proposal for a network of car-free corridors that facilitated the movement of residents, especially in areas of the city center.

Green Network and Open Space

Agriculture and horticulture have a significant role in the economy of the Latakia Governorate, which is evident in the fact that this industry contributes 22% of the total gross domestic product of the city [44]. However, these activities face significant obstacles, such as the expansion of urban areas consuming farmland and the decline in agricultural profitability, which brought the percentage of persons employed in agriculture down to 11.5% in 2011, whereas in the past it had been at 22.5% [47]. Table 3 shows the Land use and the GDP for latakia Governorate.

**Table 3.** Land use and GDP in Latakia governorate. Source: [44,47].

| Current Land Use. Source: Author Based on [47] | | Latakia Governorate GDP by Sector. Source: [44] | |
|---|---|---|---|
| Agriculture land | 43.5% | Agriculture land and forestry | 22% |
| Open and green spaces | 4% | Industry | 17% |
| Residential areas | 21.49% | Building and construction | 2% |
| Tourism | 4.95% | Hotel and tourism | 15% |
| Public services | 16.21% | Transportation and communications | 15% |
| Port area | 3.34% | Finance, insurance and real estate | 5% |
| Industrial areas | 3.39% | Public services | 24% |
| Vacant spaces | 2.16% | | |

The lack of green spaces, gardens, and open areas within the urban fabric of the city is notably conspicuous on the cartographic representation, particularly within the city center (Figure 20), where the highest population density is observed. Notably absent from the urban landscape is a public park, a crucial component integral to the local ecosystem. Furthermore, the city's primary coastal expanse is dominated by the presence of the port, impeding direct access to the shoreline. In response to these spatial limitations, Latakia Sports City, established in 1987 as the venue for the Mediterranean Sport Games, has emerged as a prominent refuge for urban residents, offering a convergence of sports facilities, expansive green spaces, and open areas. This enclave has garnered substantial popularity among city dwellers seeking respite from the pressures of urban living.

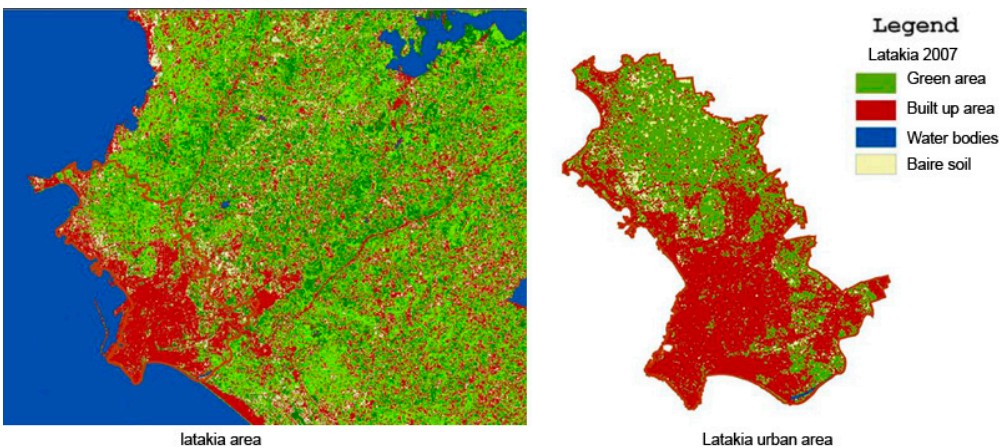

**Figure 20.** Green & built-up areas in Latakia. Source: [45].

In recent years, Latakia city has seen significant climatic changes, including changes in precipitation rates, an increase in average temperatures, a drop in groundwater levels, and other significant climatic shifts that have had an effect on the ecosystem [61–63]. The land use scheme that was suggested in the master plan did not give priority to expanding the effectiveness of the green network in the city, which plays an important role in responding

to the effects of climate change. Instead, it made bigger areas available for residential and tourism-related purposes, some of which spread over large parts of the beach. In addition to this, it decreased the amount of land used for agriculture, which is regarded to be a fundamental component of the city's local economy. Figure 21 shows the land use proposed in Latakia master plan 2008.

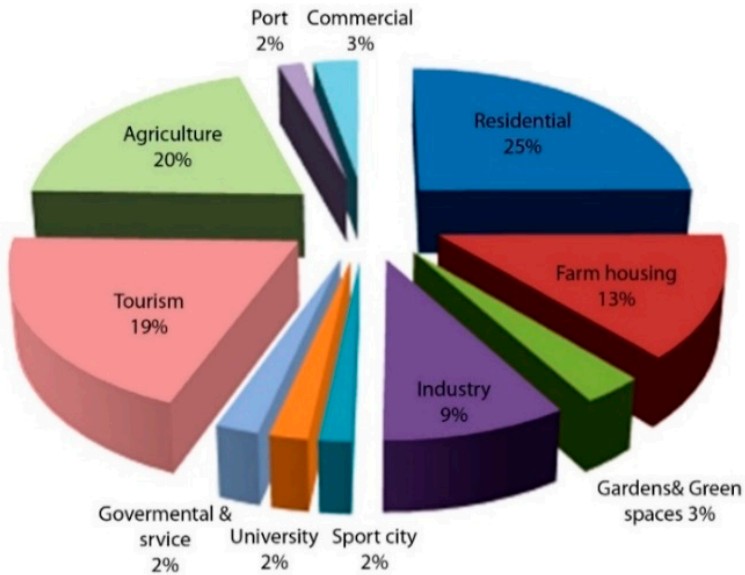

**Figure 21.** Proposed land use scheme in Latakia Master Plan 2008. Source: [57].

## 4. Discussion

Following this analysis of the urban development strategies in the three master plans, the findings are synthesized and presented in a comparison table. Table 4 includes six critical criteria, (general vision, infrastructure and services, green network, urban mobility, housing and master plan revision, and planning process), and it highlights the main features and characteristics of each master plan, providing a comprehensive overview of their similarities and differences in terms of these key criteria.

In our comparative analysis, several significant insights emerge from the examination of the master plans for Barcelona, Montpellier, and Latakia. Notably, while these plans share a common focus on land-use planning and considerations encompassing transportation, housing, and economic development, they also reveal distinct characteristics that merit careful consideration for accurate comparisons.

Barcelona and Montpellier's master plans stand as exemplars of comprehensive urban development strategies, prioritizing sustainability, equity, and residents' quality of life. These visionary plans are underpinned by a range of strategic approaches, including the enhancement of green networks, expansion of public transportation systems, promotion of mixed-use development, rehabilitation of existing infrastructure, and the cultivation of social cohesion. Such multifaceted strategies contribute to an intricate urban fabric that harmonizes societal needs and environmental preservation.

In contrast, the Latakia master plan, while emphasizing population growth, tourism, and industrial expansion, falls short in addressing sustainable development. Its centralization of vision and decision-making authority within the city council underscores a lack of substantial community involvement. This disparity is evident in the significant number of objections raised during the public submission phase (12,000 objections) [57], reflecting concerns about the plan's alignment with community aspirations. Therefore, the governorate council created a committee to investigate the complaints. The committee advocated various changes, including the cancellation of the ring road and several of the master plan's projected expansion areas.

**Table 4.** Comparison table for Latakia, Barcelona, and Montpellier master plans.

| Criteria | Barcelona | Montpellier | Latakia |
|---|---|---|---|
| General vision | Barcelona's master plan seeks to create a sustainable, inclusive, and livable city that works for everyone. Its goals include promoting economic development, improving mobility options, preserving cultural heritage, and reducing greenhouse gas emissions [49]. | Montpellier's master plan aims to create a vibrant, attractive, and competitive city center that is economically and socially inclusive. Its goals include attracting new businesses and industries, promoting social inclusion, preserving natural habitats, and improving public transportation [56]. | Latakia's master plan has a strong emphasis on promoting population growth, attracting tourists, and fostering economic development in the city. Its vision is to position Latakia as a key economic and cultural hub in the region [58]. |
| Infrastructure and services | The master plan aims to improve the quality and availability of public services, including health, education, and social services, as well as upgrading the city's infrastructure to meet the needs of its growing population [49]. | Comprehensive plan for the development of infrastructure and services, including the creation of new cultural and recreational facilities, as well [56]. | Latakia's infrastructure is less developed than the other two cities. The master plan has plans to improve infrastructure in recent years, including upgrading the port and building new roads, and new maritime roads, but more needs to be undertaken to improve access to basic services for residents [58]. |
| Urbanmobility | The city has an extensive public transportation system including metro, bus, tram, and bike-sharing networks. The master plan implements policies to reduce car usage and promote sustainable mobility, including low-emission zones and pedestrianization of city center areas [52]. | The city has a relatively small public transportation system with bus and tram networks. The master plan emphasizes improving existing transportation infrastructure rather than expanding it. It also implements policies to promote sustainable mobility, including bike-sharing programs and pedestrianization of city center areas [54]. | The city has a relatively limited public transportation system with bus networks. The master plan recognizes the need to improve transportation infrastructure and expand public transportation options, but it does not prioritize sustainable modes of transportation, such as cycling or increasing pedestrian paths. Instead, the plan focuses more on traditional modes of transportation, with more areas for parking [58]. |
| Green and Open spaces | The master plan includes a comprehensive green infrastructure plan to improve the city's biodiversity and ecological connectivity.<br><br>- Focuses on creating green corridors, parks, and public spaces to promote sustainable mobility and recreation.<br>- Has policies to encourage urban agriculture and green roofs. | The master plan includes a green network plan to improve the city's ecological connectivity, and it emphasizes preserving and enhancing existing green spaces rather than creating new ones. Furthermore, the master plan includes policies to promote urban agriculture and green roofs [56]. | The master plan does not include a comprehensive strategy to improve the green network and open spaces. This is evidenced by the fact that the plan allocates more areas for urban expansion without prioritizing the preservation or creation of green space. |
| Housing Strategy | Strong focus on social and affordable housing, with more specific and comprehensive strategy for achieving its goals in this area, including a higher target for social housing and requirements for developers to include social housing units in new developments | Strong focus on social and affordable housing with emphasis on creating mixed-income neighborhoods. Has stronger policies for promoting sustainable building practice. | The Latakia master plan primarily focuses on the introduction of new residential areas, with less emphasis on enhancing existing residential areas and providing social housing. |

**Table 4.** *Cont.*

| Criteria | Barcelona | Montpellier | Latakia |
|---|---|---|---|
| Master plan updates and revisions | The master plan undergoes periodic revisions approximately every 5–10 years, and incorporates various planning documents, such as SMPB, BCN, PEMB, and PMU, which are subject to ongoing progress updates. This process ensures that the master plan remains up-to-date and responsive to the changing demands of the city. | The master plan is revised approximately every 10 years, and it is created through a series of planning documents, such as PLH, PLU, and PADD, making the master plan up-to-date and responsive to the changing demands of the city | The master plan in Latakia is characterized by infrequent updates, occurring approximately every 25–30 years. This extended interval between revisions means that the master plan may struggle to keep pace with the ongoing changes and developments in the city, making it less efficient compared to master plans that undergo more frequent updates. |

A critical divergence in the planning processes becomes apparent when considering the frameworks utilized in each city. Barcelona and Montpellier adopt a decentralized approach, leveraging a diverse array of planning instruments, such as BCN, PEMB, and PMU (Barcelona, Spain), and SCOT, PLU, and PADD (Montpellier, France). This inclusive methodology fosters a comprehensive vision of urban development, with each document centering on distinct aspects of urban life. On the contrary, the Latakia master plan reflects a more centralized process, driven by government influence and technical expertise. This approach, while efficient, lacks the community engagement necessary for a holistic and sustainable urban vision.

Furthermore, the frequency of master plan revisions reveals another facet of divergence. Barcelona and Montpellier engage in periodic updates, ensuring alignment with evolving urban dynamics approximately every decade. This flexibility allows these cities to effectively adapt to changing circumstances and address emerging challenges. In contrast, Latakia's less frequent revisions, occurring every 20 to 25 years, may hinder its ability to optimally accommodate the city's dynamic transformations over time.

## 5. Conclusions

In conclusion, this comparative analysis of master plans for Barcelona, Montpellier, and Latakia underscores the importance of tailored urban development strategies that balance diverse needs and aspirations. Barcelona and Montpellier's comprehensive approaches, underpinned by decentralized planning frameworks and frequent revisions, offer valuable insights into achieving sustainable, equitable, and vibrant urban environments.

For Latakia, elevating the effectiveness of the master plan demands a shift toward sustainable development that harmonizes land use, environmental preservation, and holistic economic advancement. The pivotal agricultural sector can play a transformative role, driving economic prosperity and community well-being through strategic empowerment of farmers. Drawing inspiration from successful green network and urban agriculture strategies can further enhance Latakia's ecological sustainability and overall quality of life.

Moreover, embracing diverse strategies, such as urban renewal projects, fosters balanced and equitable urban development, revitalizing underdeveloped areas and enhancing overall urban resilience. The incorporation of cooperative and social housing initiatives, mirroring successful approaches such as those of Montpellier, enhances inclusivity and addresses urban housing demands.

To bolster urban resilience against climate change, expanding green spaces and promoting non-motorized transportation modes emerge as vital strategies. These elements counteract heat effects, enhance air quality, and provide essential recreational spaces while mitigating carbon emissions and traffic congestion.

Finally, crafting a sophisticated master plan requires rigorous research, collaboration with experts across various fields, and data-driven decision-making. By incorporating best

practices and international standards, a well-informed and comprehensive plan can effectively guide Latakia's urban development, creating a resilient, sustainable, and prosperous future for its residents.

**Author Contributions:** Conceptualization, N.K., M.S. and A.F.; methodology, N.K. and M.S.; software, N.K.; validation, N.K., A.F. and M.S.; formal analysis, N.K, A.F. and M.S.; investigation, N.K.; resources, N.K., A.F. and M.S.; data curation, N.K.; writing—original draft preparation, N.K., M.S. and A.F.; writing—review and editing, A.F. and M.S.; visualization, N.K.; supervision, A.F. and M.S.; project administration, A.F. and M.S. All authors have read and agreed to the published version of the manuscript.

**Funding:** This research received no external funding. The publication was supported by the Hungarian University of Life Sciences—MATE.

**Data Availability Statement:** Not applicable.

**Acknowledgments:** We wish to thank the work of the anonymous reviewers.

**Conflicts of Interest:** The authors declare no conflict of interest.

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
