# Peer review of "The Role of the Master Plan in City Development, Latakia Master Plan in an International Context"

_land, doi:10.3390/land12081634_

Round 1

Reviewer 1 Report

Dear authors,

Thank you for allowing me to read the manuscript "The role of Master Plan in city development, Latakia Master plan in an international context".

The theme presented is extremely relevant nowadays, and the approach presented by the authors is important as a "benchmarking" of good examples to be applied in other cities/regions, especially in developing countries.

The theme fits within the scope of the journal. Despite the interest of the theme and the reflection presented, the manuscript does not fully have the structure of a scientific article and is sometimes too descriptive (although there is an effort of comparative synthesis by the authors at the end).

Substantially, it is suggested that the authors reinforce the literature review component, revise section 2. Materials and Methods as some aspects does not fit here, convert section 3. discussion into results, convert section 4. conclusions into discussion and create a small section of conclusion OR convert to discussion and conclusions. Given the highly descriptive nature of the document, it is important that the reader, planner or decision-maker, be able to quickly absorb the aspects considered as good practices (possibly to be included in the conclusion as bullet points).

I let some specific notes to improve the manuscript:

- Considering that you are submiting the manuscript to Land, it could be intereting to reinforce the literature review about the land/city evolution in the European cities.

- The authors identify in the section 2. Materials and methods a set of bibliography, but indeed, the references are not discussed, but only agrouped by theme. These references should be explored in a literature review section that do not exist. Indeed, the documents 43-48 are not the theoretical bases if this research but the objects of analysis.

- section 3. shoud be results and not discussion yet

- is there any updated data about Barcelona`s Census?

- clarify when the Barcelona Vision 2020 appeared

- please, clarify in the several referred documents the published date and the vigent period

- please, add scales in the various maps of figure 11

- please, check if all the figures are cited in the text

- check overpositioning of figure 18 title and legend

- as the comparative table is a result of your work, you should consider to put it as a subsection of RESULT section or create a DISCUSSION section with the table and its discussion, and a new section of CONCLUSIONS 

- despite the very interesting and comparative discussion,  the authors could save the highlights about what should be considered in Latakia planning to the conclusions, informing in a clear way the decision makers (of Latakia and abroad).

Best regards.

Reviewer 2 Report

The authors study a Latakia (Syria) Master Plan, comparing it with master plans in two other Mediterranean and coastal cities, Barcelona (Spain) and Montpellier (France). The role of a Master Plan in city development is always an actual and significant research topic. The theme of this paper is interesting, but at this stage, its presentation does not allow to evaluate the work entirely. Some issues need to be revised, which significantly affect the quality of the article and can only be considered for publication after careful revision. The specific issues are as follows:

- The Introduction does not describe the research field’s current state. The literature review must be improved, and some questions need to be explained emphatically. The first issue you need to tackle is the concept of a “master plan”. You provide some explanation in lines 44-56, but it is not clear whether you refer to master (physical) planning on some other types of master plans. This must be clarified, and a definition of a master plan needs to be provided.

- In your research, you are comparing five master plans of Barcelona, one master plan of Latakia, and several (?) master plans from Montpellier, all of them of various scopes and types. For reliable results, you must compare the same level and type of master plans in selected cities.

- The critical information about analysed documents are missing and must be included, such as the exact title, year of development/adoption, scope, etc. Is the Master Plan of Latakia an officially adopted document? Did all alternatives of the MP given in Table 2 get approved? When you mention a plan in the text, please specify exactly which one you are discussing (for example, lines 228, 274, 408…).

- Fourth, and not unrelated, you specify the elements of the master plan for comparison – general vision, housing policies, urban mobility and greenery network. However, you are using information from various plans to compare an element. For example, in Barcelona section “3.1.3.1. Housing development strategy”, some regeneration projects are presented; in Montpellier, it is a local housing program, and only in Latakia is it an element of the master plan. Please be consequential in the analysis and compare the same-level plan elements.

- Line 132. Section “3. Discussion” should read as 3. Results

- Section .”4 Conclusions” should read as 4. Discussion and conclusions. In this section, reference support is required

- Table 4. How can you compare the planning process and decision-making in three cities when you have not previously analysed and discussed this element? Please correct this by deleting this from the table or adding the needed analysis. In addition, you have mismatched information in this table for Barcelona and Montpellier.

- Table 3 “Source [35]” should read as Source [45]

- Figures 3, 5, 7, 8, 12, 15, 19, and 20 are blurry

Reviewer 3 Report

The research requires a scientific robustness. 

It is not clear why these three case studies are selected - and compared. As the masterplans were developed in extremely different socio-economic and cultural contexts, with different GDPs and sources of economic potential on national levels .. The paper now present very general statements that are results of very general contexts and legislative frameworks within which these plans had been developed.

In order to improve the article, there can be more ways taken. For instance, maybe focus on more particular details can bring in more interest - such as comparison of using different indexes / and their results. Or, develop a robust methodology that would allow to compare the vision of the masterplans (EU countries that are  under the same legislative system) ... 

Round 2

Reviewer 1 Report

Dear authors,

Dear editor,

Thank you for your effords to improve the manuscript. This version accomplish all my requests. In this sense, I consider that this version as cnow conditions to be published.

Only a very small note to correct: in the lines 47-53, as this is a direct citation, you should put the pages.

Best regards

Reviewer 3 Report

From the revised methodology it is still not clear how and why the case study were selected. This ambiguity opens up the question if the chosen selection is bringing any research validity. In order to make research adequately robust, the recommendation is to clearly state the selection criteria that are defined - such as: similar  demographic trajectory, or similar economic intervention situation (which for instance olympic games are, as they generate very specific needs for the mobility, accessibility, temporary uses of built area, etc.. Thus it would make more robustness to compare urban plans of the cities that have had to accommodate the Olympic Games.).  Therefore, the current state of the article still does not make it very clear why to choose Barcelona (1.62 mil populations), Montepellier (277, 000 population), Lakatia (cca 700, 000 population). Why not to choose the cities of the same population and the cities with the similar main challenges to address? The same for the other aspects of the selection criterions. 
